# Study on Rh(I)/Ru(III) Bimetallic Catalyst Catalyzed Carbonylation of Methanol to Acetic Acid

**DOI:** 10.3390/ma13184026

**Published:** 2020-09-11

**Authors:** Shasha Zhang, Wenxin Ji, Ning Feng, Liping Lan, Yuanyuan Li, Yulong Ma

**Affiliations:** 1College of Chemistry and Chemical Engineering, Ningxia University, Yinchuan 750021, China; zssycdi@163.com (S.Z.); 18408610963@189.cn (L.L.); liyy@nxu.edu.cn (Y.L.); nxylma@163.com (Y.M.); 2State Key Laboratory of High-efficiency Utilization of Coal and Green Chemical Engineering, Ningxia University, Yinchuan 750021, China; 17854170389@139.com

**Keywords:** carbonylation, acetic acid, bimetallic catalyst

## Abstract

In this study, a Rh(I)/Ru(III) catalyst with a bimetallic space structure was designed and synthesized. The interaction between the metals of the bimetallic catalyst and the structure of the bridged dimer can effectively reduce the steric hindrance effect and help speed up the reaction rate while ensuring the stability of the catalyst. X-ray photoelectron spectroscopy (XPS) results show that rhodium accepts electrons from chlorine, thereby increasing the electron-rich nature of rhodium and improving the catalytic activity. This promotes the nucleophilic reaction of the catalyst with methyl iodide and reduces the reaction energy barrier. The methanol carbonylation performance of the Rh/Ru catalyst was evaluated, and the results show that the conversion rate of methyl acetate and the yield of acetic acid are 96.0% under certain conditions. Furthermore, during the catalysis, no precipitate is formed and the amount of water is greatly reduced. It can be seen that the catalyst has good stability and activity.

## 1. Introduction

As an important organic raw material, acetic acid is widely used in chemical, pharmaceutical, pesticidal, and food industries, among others. During the early 1990s, BP Chemicals developed an iridium complex catalyst that was in competition with the rhodium complex catalyst, and the new technology was commercialized as the Cativa process in 1995. The Monsanto process was commercialized in 1966 using an improved Rh-based homogeneous catalyst for a liquid-phase methanol carbonylation with a methyl iodide promoted rhodium catalyst [1,2,3]. It is an important raw material for the synthesis of a vinyl acetate monomer and acetic anhydride. At present, Monsanto’s methanol carbonylation to acetic acid process with (Rh(CO)_2_I_2_)^−^ as the catalytic active center has become the mainstream process for global acetic acid production due to mild reaction conditions and sufficient raw material sources [4,5,6,7]. In the Monsanto rhodium/iodine catalyst catalyzed carbonylation to acetic acid reaction, Forster [8] proposed that the reaction mainly undergoes four elementary reactions: CH_3_I oxidative addition, ligand migration, CO coordination, and CH_3_COI reduction. Among them, the first step in the CH_3_I oxidation addition reaction is the rate-determining step in the entire catalytic process [9]; therefore, to improve the activity of the catalyst, it is necessary to reduce the reaction barrier of the CH_3_I oxidation addition.

However, Rh(I) is unstable during the reaction of the Monsanto rhodium/iodine catalyst. When the CO partial pressure is low, (Rh(CO)_2_I_2_)^−^ interacts with hydroiodic acid to form dimers to form Rh(III) precipitates [10,11,12,13], causing the deactivation of the precious metal catalyst. Therefore, a large number of polar solvents such as H_2_O and acetic acid are added to the system to increase the solubility of the catalyst, thereby improving the stability of the catalyst. In the industrial processing of methanol to acetic acid, water and acetic acid are added to the system to prevent the catalyst from deactivating. Therefore, under the premise of ensuring the activity of the catalyst, selecting a highly stable catalyst will effectively increase the production efficiency of acetic acid and reduce the production cost, which has also become a research hotspot in the field of carbonylation to acetic acid.

Bimetallic complex catalysts with heteronuclear structures are often used to show better performance than single-metal catalysts through metal–metal interactions and their ligand effects [14,15,16]. As a result, the catalyst has become a research hotspot [17,18,19,20,21,22]. The bridged ligand and Rh coordinate to form a bidentate coordination structure, and after the ligand is coordinated with Rh, it can firmly grasp Rh. The bimetallic complex with a stable structure can use the steric effect to prevent catalyst deactivation.

A catalyst with a bimetallic structure (Rh(I)/Ru(III)) was designed and synthesized. The catalytic evaluation of catalysts using methanol and CO as raw materials for carbonylation to produce acetic acid was compared with the (Rh(CO)_2_I_2_)^−^ catalyst in the Monsanto process. Through theoretical calculations, the structural characteristics and the catalytic reaction mechanism of Rh(I)/Ru(III) were studied at the molecular level, and the internal relationship between the catalyst structure and catalytic performance was discussed [23,24]. The Rh(I)/Ru(III) catalyst was synthesized under theoretical guidance, and its structure was confirmed through characterization. The performance of the catalyst for the preparation of acetic acid by methanol carbonylation was evaluated. This provides new ideas for the design of carbonylation catalysts.

## 2. Materials and Methods

### 2.1. Raw Materials and Reagents

Methanol (CH_3_OH), acetic acid (CH_3_COOH), and sodium carbonate (Na_2_CO_3_) were analytical-grade products from Shanghai Aladdin Reagent Co. Ltd. (Shanghai, China). Lithium iodide (LiI) and potassium iodide (KI) were analytical grade products from Shanghai Macklin Biochemical Technology Co. Ltd. (Shanghai, China). Methyl iodide (CH_3_I) was an analytical-grade product from Shandong West Asia Chemical Co. Ltd. (Linyi, China). Rhodium chloride hydrate (RhCl_3_·3H_2_O) and ruthenium chloride hydrate (RuCl_3_·3H_2_O) were both analytical grade products from Shanxi Dongwal Chemical Co. Ltd. (Taiyuan, China). Carbon monoxide (CO) and nitrogen (N_2_), with a volume fraction of 99.999%, were both from the Ningxia Guangli Gas Plant (Yinchuan, China).

### 2.2. Theoretical Calculation Method

Based on Lanl2dz, the mechanism of methanol carbonylation of the catalyst was studied theoretically using density functional theory (DFT) and the B3LYP method. At the molecular level, the catalytic reaction mechanism was studied and the structure and energy of intermediates and transition states were obtained [25]. All energies were corrected at zero, while the reaction energy was compared with the catalytic performance of the Monsanto iodine catalyst. All calculations were performed using the Gaussian 09 quantum chemistry program.

### 2.3. Catalyst Preparation

#### 2.3.1. Synthesis of Precursor Rh_2_(CO)_4_Cl_2_

To begin, 0.2 g of rhodium chloride hydrate (RhCl_3_·3H_2_O) was weighed, and the particles were ground into to a powder state. The powder was transferred to a U-shaped glass tube with a sand core device at the bottom, in which the air had been removed. Then, 0.5 MPa of CO passed, and the tube was allowed to react in an oil bath at a constant temperature of 80–90 °C for 6 h. As the reaction progressed, orange-red crystals appeared in the glass tube. After the reaction was over, the orange-red crystals were collected into a brown reagent bottle where they were weighed and stored to dry.

#### 2.3.2. Synthesis of Rh(I)/Ru(III) Bimetallic Catalyst

Rh_2_(CO)_4_Cl_2_ and RuCl_3_·3H_2_O were weighed at a molar ratio of 1:2 and dissolved in 10 mL of distilled water. This solution was mixed by stirring at room temperature for 30 min, and the mixture was then slowly added dropwise to 50 mL of 0.1 mol/L Na_2_CO_3_ solution. The resulting mixture was placed in a water bath at a constant temperature of 50 °C and reacted for 10 h until black precipitate appeared. After cooling, centrifuging, and drying the mixture at 50 °C, black solid particles were obtained.

#### 2.3.3. Evaluation of Catalyst Performance

The total mass of reactants was 25 g. According to the H_2_O:CH_3_OH:LiI:CH_3_COOH:CH_3_I mass ratio of 6:24:4:54:12, H_2_O, CH_3_OH, LiI, CH_3_COOH, and CH_3_I were added to a 100 mL autoclave. Then, 0.03 g of the Rh(I)/Ru(III) bimetallic catalyst was added. The initial CO pressure was 3.5 MPa, and the reaction temperature was 190 °C. The stirring rate was 500 r/min, and the reaction time was 60 min. After the reaction was completed, a sample was taken for product analysis.

### 2.4. Characterization of the Catalyst

X-ray photoelectron spectroscopy (XPS) measurements were performed using a Thermo Fisher ESCALAB 250 Xi (Symerfish Technologies, New York, NY, USA) equipped with an Al Ka radiation X-ray source. FTIR measurements were carried out by an Alpha FTIR spectrometer (Brucker Optics, Ettlingen, Germany) equipped with exchangeable sampling modules. The spectra were collected as the average of at least 200 scans at a resolution of 4 cm^−1^ in the frequency range of 2500–400 cm^−1^. Inductively coupled plasma optical emission spectroscopy (ICP-OES) experiments were performed on an Agilent 725 optical emission spectrometer (Agilent Technology Limited, Santa Clara, CA, USA). Quantification was performed by an external standard method.

### 2.5. Catalyst Performance Analysis Method

The product was quantitatively analyzed using a Shimadzu gas chromatograph (GC-2014), an Intercap FFAP polar capillary column, and a hydrogen flame ionization detector (FID). The injection volume was 1 μL, and the heating rate was 5 °C/min from 32 to 60 °C. This was held for 2 min, then the temperature was increased to 150 °C at a 10 °C/min heating rate and held for 2 min. Finally, the product was quantified by an external standard method. The qualitative analysis was performed using a Shimadzu gas chromatograph mass spectrometer (GCMS-QP2010, Shimadzu Production Centre, Kyoto, Japan) under the same conditions. Taking the conversion rate of methanol (*x*, %), the selectivity of acetic acid (*s_HAc_*, %), the selectivity of methyl acetate (*s_MeOAc_*, %), the yield of acetic acid (*y_HAc_*, %) and the yield of methyl acetate (*y_MeOAc_*, %) is used as the evaluation index of catalyst catalytic performance, and the calculation formula is as follows:$$x=\frac{n_{2}}{n_{1}}\times 100\%$$
$$s_{HAC}=\frac{n_{3}}{n_{2}}\times 100\%$$
$$s_{MeOAc}=\frac{n_{4}}{n_{2}}\times 100\%$$
$$y_{HAC}=x\times s_{HAc}\times 100\%$$
$$y_{MeOAc}=x\times s_{MeOAc}\times 100\%$$
$$TOF=\frac{reaction\;times}{\left(number\;of\;active\;sites\times \;reaction\;time\right)}$$
$$TON=\frac{\;reaction\;times}{number\;of\;active\;sites}$$
where *n*_1_ is the amount of methanol added, *n*_2_ is the amount of methanol consumed, *n*_3_ is the amount of acetic acid produced, and *n*_4_ is the amount of methyl acetate produced; the unit is mol.

The product was qualitatively analyzed by GC-MS. Figure 1 shows that the peaks of the spectrum corresponded to methyl iodide, methyl acetate, methanol, and acetic acid.

## 3. Results

### 3.1. Molecular Design

It can be seen from Figure 2d that Rh(I)/Ru(III) is a four-coordinate bimetallic space structure, and the distance between the central metal atom Rh and Ru is 3.5846 Å. The dimer usually has the most stable planar structure and lowest energy. We assume the planar structure of the Rh/Ru catalyst. The calculation results show that the distance between Rh and Ru in the planar structure is 3.6832 Å, and the molecular energy of the space structure is 54.67 kJ/mol lower than that of the planar structure. This reduced energy mainly comes from the interaction between Rh and Ru, thereby improving the stability of the catalyst.

### 3.2. Catalytic Reaction Mechanism

The mechanism of Rh(I)/Ru(III) catalyzing methanol carbonylation is similar to that of the Monsanto process (see Figure 3a). Figure 4 shows the geometric structure of reactants, transition states, intermediates, and products in methanol carbonylation. As shown in Figure 3 and Figure 4, during the Rh(I)/Ru(III) catalysis process, the oxidation addition reaction of CH_3_I passes through the transition state (TS1), while the Rh(I)/Ru(III) bimetallic catalyst catalytic reaction is from the original four-coordinate structure to a stable six-coordinate four-pyramid double-cone structure (TN1). During the oxidative addition of CH_3_I, as shown in Figure 2d, the Rh/Ru catalyst was obtained through structural optimization. The bond length of Cl(1)/Rh (2.5141 Å) is longer than that of Cl(2)/Rh (2.5065 Å); the bond energy is also lower and easier to extend. As shown in Figure 4, the Rh/Cl(1) bond was first elongated (from 2.5141 to 3.0090 Å) and then recovered. The spatial structure of the catalyst was slightly deformed. It can be seen that this deformation can cooperate with the addition of the rate-determining step CH_3_I, effectively reducing the energy barrier of the rate determination reaction and thereby improving the activity of the catalyst. After carbonyl migration and carbonyl coordination, it is finally eliminated by CH_3_COI reduction and hydrolysis to form acetic acid, thus completing the entire catalytic cycle process.

It can be seen from Figure 5 that compared with the Monsanto catalyst, the CH_3_I oxidation reaction of methanol carbonylation catalyzed by the Rh(I)/Ru(III) bimetallic catalyst reduces the energy barrier by 23.88 kJ/mol. At this time, the elimination reaction has become the rate-determining step. Compared with Monsanto’s rhodium iodide catalyst, the Rh(I)/Ru(III) bimetallic catalyst has a higher stability due to the interaction between Rh/Ru metal and its bidentate structure. In the process of ligand migration, the Rh/Cl (1) bond is elongated, but Cl (2) and Rh are still firmly grasped to prevent the decomposition and inactivation of the precipitate. As a result, it is advantageous to increase the rate of the carbonylation reaction and the selectivity and activity of the catalyst.

Figure 6 shows a comparison of the infrared image of Rh_2_(CO)_4_Cl_2_ with the infrared image of RhCl_3_·3H_2_O. At 2007 and 1900 cm^−1^, there are two rhodium terminal carbonyl peaks. It can be seen that Rh_2_(CO)_4_Cl_2_ was successfully synthesized. In the infrared spectrum (Figure 6), the characteristic absorption peaks of the terminal carbonyl group of the Rh(I)/Ru(III) catalyst appear at 1600 and 1700 cm^−1^. Comparing the infrared images of Rh_2_(CO)_4_Cl_2_, it can be seen that due to the influence of the ligand, the peak of the terminal carbonyl rhodium in the Rh(I)/Ru(III) bimetallic catalyst has a red shift. At the same time, due to the influence of the central metal Rh, its vibration frequency is significantly reduced.

Figure 7 shows the XPS spectrum of the Rh(I)/Ru(III) bimetallic catalyst and precursor. XPS requires calibration since charging of the sample will influence the kinetic energies. Spectra from samples have been charge-corrected to give the adventitious C1s spectral component (C–C, C–H) a binding energy of 284.8 eV. This process has an associated error of ± 0.1–0.2 eV [26]. As shown in Figure 7, compared with Rh3d5/2 (308.9 eV) in Rh_2_(CO)_4_Cl_2_, the binding energy (308.5 eV) of the Rh(I)/Ru(III) bimetallic catalyst is reduced by 0.4 eV, while the binding energy of Ru3d3/2 (284.4 eV) is reduced by 2 eV compared to RuCl_3_/3H_2_O and Ru3d3/2 (286.4 eV). The Cl2p in the Rh(I)/Ru(III) bimetallic catalyst is 199.7 eV. After coordination with Rh and Ru, the binding energy of Cl2p in Rh_2_(CO)_4_Cl_2_ and RuCl_3_/3H_2_O increased by 1.2 eV and 1.1 eV, respectively. It can be seen that during the coordination process, a Cl/Rh coordination bond and a Cl/Ru coordination bond are formed. The binding energy of rhodium and ruthenium is reduced, and the binding energy of chlorine is increased, indicating that the electrons of chlorine are transferred to rhodium and ruthenium. The electron-rich nature of rhodium increases the activity of the catalyst and increases the reaction rate.

The composition of the Rh(I)/Ru(III) bimetallic catalyst was investigated using inductively coupled plasma optical emission spectrometry (ICP-OES, AGILENT 725). The mole ratio of Rh/Ru was 1.15. Based on the characterization results of XPS and ICP-OES, we believe that the experimentally obtained Rh(I)/Ru(III) bimetallic catalyst configuration conforms to the theoretical design configuration.

### 3.3. Comparison of Catalyst Performance

Table 1 shows the experimental results of methanol carbonylation catalyzed by Rh(I)/Ru(III) catalyst and the Monsanto catalyst under the same conditions. It can be seen from Table 1 that when Rh(I)/Ru(III) catalyzes the carbonylation reaction, its catalytic performance is significantly better than that of the Monsanto catalyst. At the end of the reaction, some precipitation of RhI_3_ appeared in the reactor in the Monsanto catalytic system, but no precipitation occurred in the Rh(I)/Ru(III) catalytic system, indicating that Rh(I)/Ru(III) has a better stability [27,28,29,30]. As a result of the catalytic performance of RuCl_3_/3H_2_O, it has no catalytic activity, but it stabilizes the catalyst structure and forms a folded structure, which reduces the overall energy of the catalyst system.

### 3.4. Catalyst Performance Evaluation

In the Monsanto catalyst, to avoid the deactivation of the catalyst active metal Rh when the CO partial pressure is low, a large amount of water is usually added. We investigated the effect of water content on the performance of the bimetallic catalyst in the methanol carbonylation reaction [31,32,33]. Figure 8 shows the performance of Rh(I)/Ru(III) bimetallic catalyst catalyzed carbonylation of methanol under different H_2_O mass fractions. Figure 8 shows that the selectivity of acetic acid first increases with increasing water content and then decreases. At a water content of 6 wt%, acetic acid has the highest selectivity, 95.90%, and no other by-products.

As can be seen from Figure 9, the selectivity of acetic acid also gradually increases with the increase of the amount of solvent. When the solvent content is 54 wt%, the acetic acid selectivity is as high as 95.90%. Considering the production and production costs of by-products, the optimal amount of solvent in this experiment was 54 wt%.

It can be seen from Figure 10 that in the Rh(I)/Ru(III) bimetallic catalytic system, the methanol conversion rate is 100.00%. With the increase of the system pressure, the selectivity of acetic acid increases first and then decreases. The selectivity of methyl acetate gradually decreases and then increases. At 3.5 MPa, the acetic acid selectivity is 95.90%. When the pressure continues to increase, by-product methyl acetate appears in the product, and the acetic acid selectivity begins to decrease. Therefore, 3.5 MPa is the optimal reaction pressure of the system.

As shown in Figure 11, with the increase of reaction temperature, the selectivity of acetic acid shows an increasing trend. At 180, 190, and 200 °C, the selectivity of acetic acid reached over 90%. At 190 °C, the highest acetic acid selectivity is 96.32%, and no other by-products are produced. Therefore, 190 °C is the optimal reaction temperature for this reaction.

## 4. Discussion

Under the guidance of theory, the Rh(I)/Ru(III) catalyst with bimetallic space structure was synthesized. Through theoretical calculations, it is found that the bimetallic catalyst with a 3D space structure has a shorter Rh/Ru spacing, forming a folded structure, reducing the system energy, and increasing the catalyst stability. In addition, the presence of steric hindrance can also prevent the deactivation of the Rh center of the catalyst. XPS results show that rhodium accepts electrons from chlorine, which promotes its electron enrichment and is beneficial to the improvement of catalyst activity. This is conducive to the nucleophilic reaction of the catalyst with methyl iodide and reduces the reaction energy barrier. The Rh/Ru bimetallic catalyst has a bridged dimer space structure. During the oxidation addition of CH_3_I, one bridge Cl seizes Rh and the other bridge Cl/Rh is elongated. After the addition, the bridge Cl/Rh bond-length is restored. This synergistic effect can reduce the reaction energy barrier of CH_3_I addition and ensure the stability of the catalyst.

A Rh(I)/Ru(III) bimetallic catalyst was synthesized, and its performance catalyzing methanol carbonylation to produce acetic acid was investigated. The Rh(I)/Ru(III) bimetallic catalyst optimizes the reaction system of methanol carbonylation to acetic acid. The results show that at a reaction temperature of 190 °C, the initial CO pressure is 3.5 MPa, the solvent is 54 wt% acetic acid, and the water content is 6 wt%. The carbonylation reaction is the best, and the selectivity of acetic acid is 96.32%. No precipitation occurs in the system, and the catalyst maintains good stability. The amount of solvent water is significantly lower than that of industrial water (~13–16 wt%), no by-products are produced, industrial production costs are reduced, and energy consumption for subsequent separation is saved.

## Figures and Tables

**Figure 1 materials-13-04026-f001:**
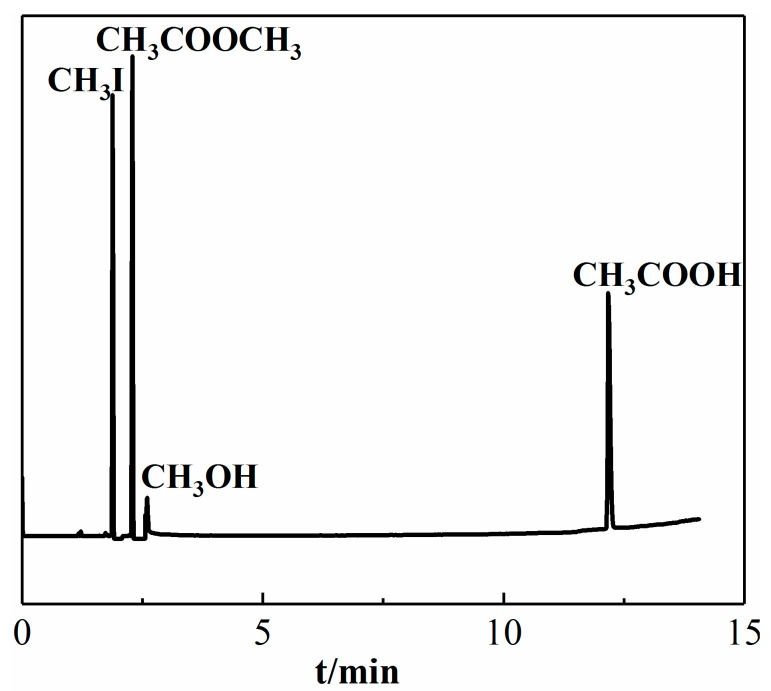
GC-MS diagram of carbonylation reaction product.

**Figure 2 materials-13-04026-f002:**
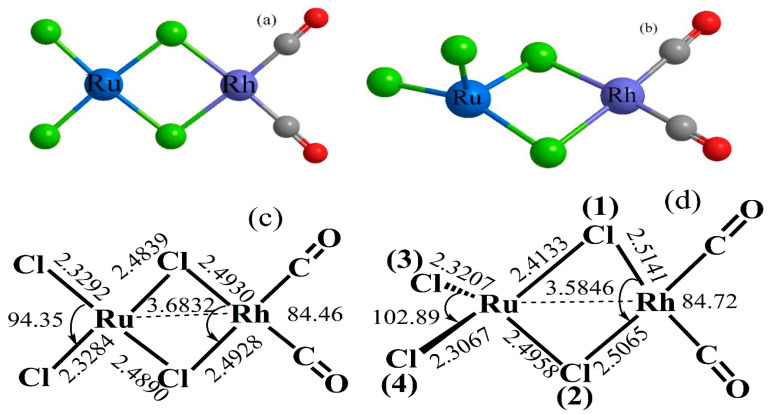
Geometric structure of Rh(I)/Ru(III) bimetallic catalysts: (**a**,**c**) plane structure of Rh(I)/Ru(III) bimetallic catalysts; (**b**,**d**) stereo structure of Rh(I)/Ru(III) bimetallic catalysts (bond length in Å, bond angle in degrees).

**Figure 3 materials-13-04026-f003:**
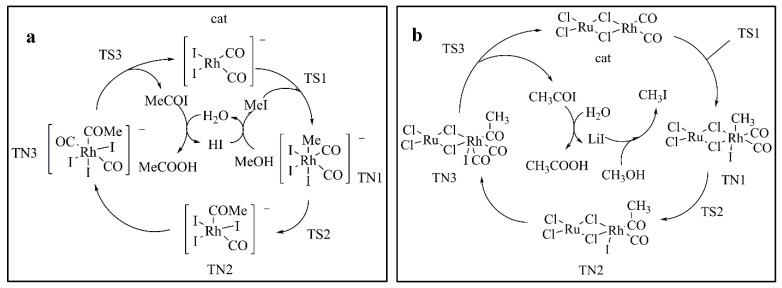
Diagram of the reaction mechanism for methanol carbonylation: (**a**) cycle for Monsanto catalyst catalyzed carbonylation of methanol to acetic acid; (**b**) cycle for Rh(I)/Ru(III) bimetallic catalyst catalyzed carbonylation of methanol to acetic acid.

**Figure 4 materials-13-04026-f004:**
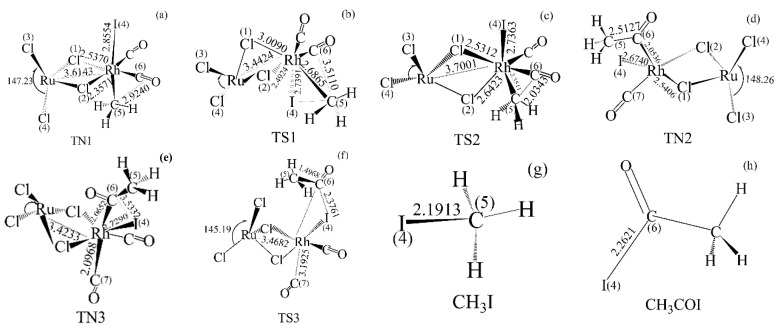
Geometric structure of reactants, transition states, intermediates, and products in methanol carbonylation (bond length in Å, bond angle in degrees): (**a**) CH_3_I addition reaction product (TN1); (**b**) CH_3_I addition reaction state (TS1); (**c**) ligand migration reaction state (TS2); (**d**) ligand migration reaction product (TN2); (**e**) CO addition reaction product (TN3); (**f**) CH_3_COI reductive elimination reaction state (TS3); (**g**) CH_3_I; (**h**) CH_3_COI; (DFT, B3LYP/Lanl2dz).

**Figure 5 materials-13-04026-f005:**
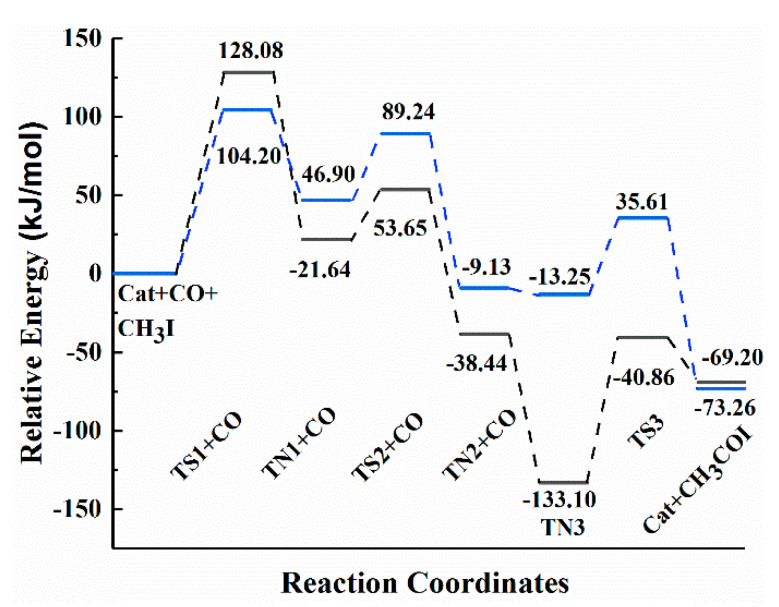
Potential energy diagram for the reaction pathway. Black line: The reaction barrier diagram of the Monsanto catalyst catalyzing methanol carbonylation to acetic acid. Blue line: The reaction barrier diagram of Rh(I)/Ru(III) bimetallic catalyst catalyzing methanol carbonylation to acetic acid.

**Figure 6 materials-13-04026-f006:**
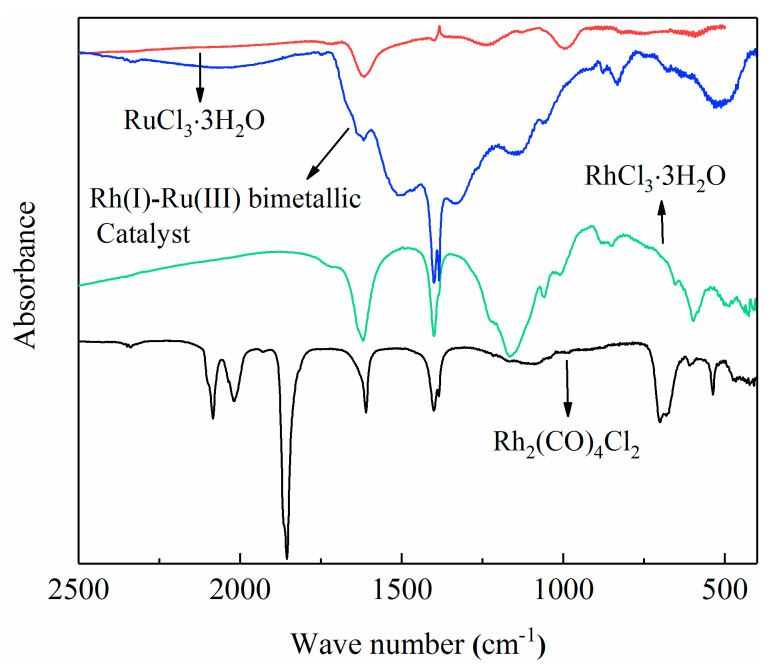
FTIR spectra of different catalysts. Blue line: FTIR spectra of Rh(I)/Ru(III) bimetallic catalyst; green line: FTIR spectra of RhCl_3_·3H_2_O; red line: FTIR spectra of RuCl_3_·3H_2_O; black line: FTIR spectra of (Rh_2_(CO)_4_Cl_2_).

**Figure 7 materials-13-04026-f007:**
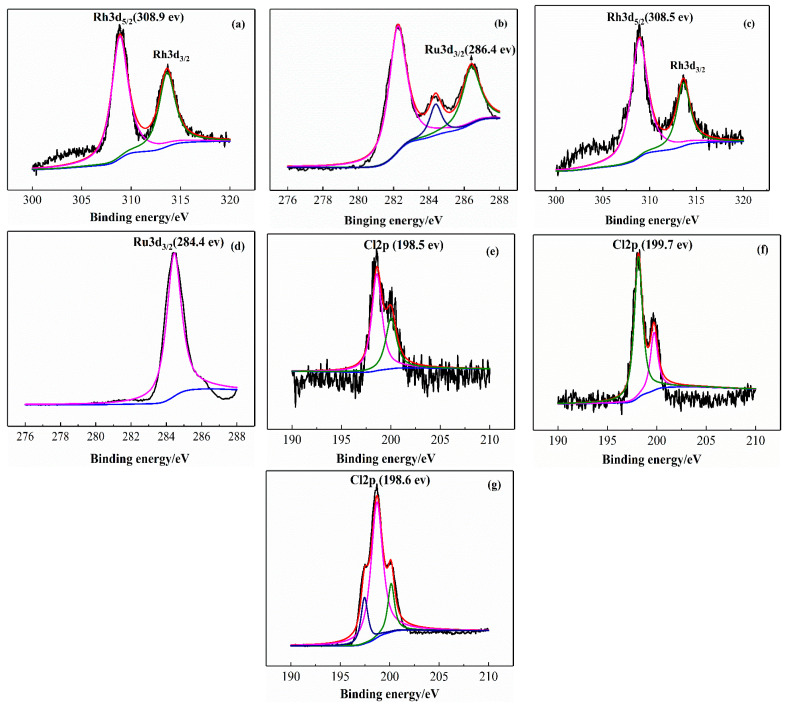
XPS spectra for Rh(I)/Ru(III) bimetallic catalyst and precursors: (**a**) XPS spectra of Rh (Rh_2_(CO)_4_Cl_2_); (**b**) XPS spectra of Ru (RuCl_3_·3H_2_O); (**c**) XPS spectra of Rh (Rh(I)/Ru(III) bimetallic catalyst); (**d**) XPS spectra of Ru (Rh(I)/Ru(III) bimetallic catalyst); (**e**) XPS spectra of Cl_2p_ (Rh_2_(CO)_4_Cl_2_); (**f**) XPS spectra of Cl_2p_ (Rh(I)/Ru(III) bimetallic catalyst); (**g**) XPS spectra of Cl_2p_ (RuCl_3_·3H_2_O).

**Figure 8 materials-13-04026-f008:**
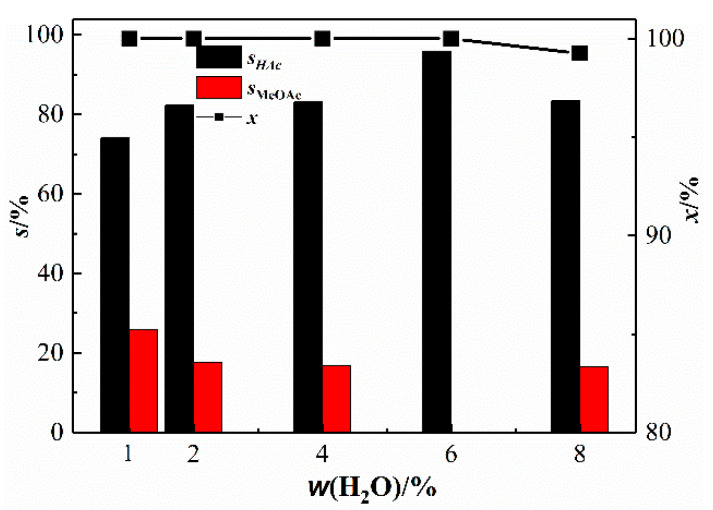
Catalytic performance of Rh(I)/Ru(III) bimetallic catalysts at different contents of water (T = 190 °C; *p* = 3.5 MPa; *t* = 60 min; w(CH_3_COOH) = 54 wt%; x/%: conversion rate of methanol).

**Figure 9 materials-13-04026-f009:**
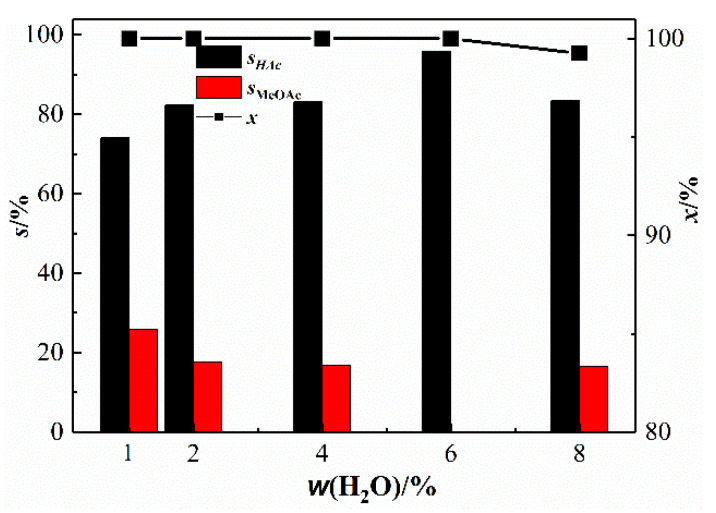
Catalytic performance of Rh(I)/Ru(III) bimetallic catalyst in different solvent contents (T = 190 °C; *p* = 3.5 MPa; *t* = 60 min; w (H_2_O) = 6 wt%; x/%: conversion rate of methanol).

**Figure 10 materials-13-04026-f010:**
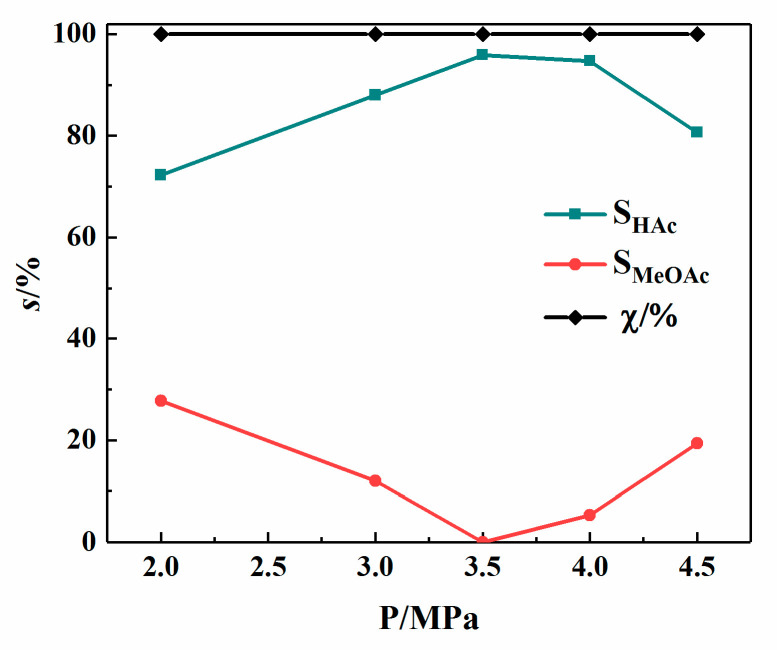
Catalytic performance of Rh(I)/Ru(III) bimetallic catalysts at different pressures (T = 190 °C; *t* = 60 min; w(CH_3_COOH) = 54 wt%; w(H_2_O) = 6 wt%; x/%: conversion rate of methanol).

**Figure 11 materials-13-04026-f011:**
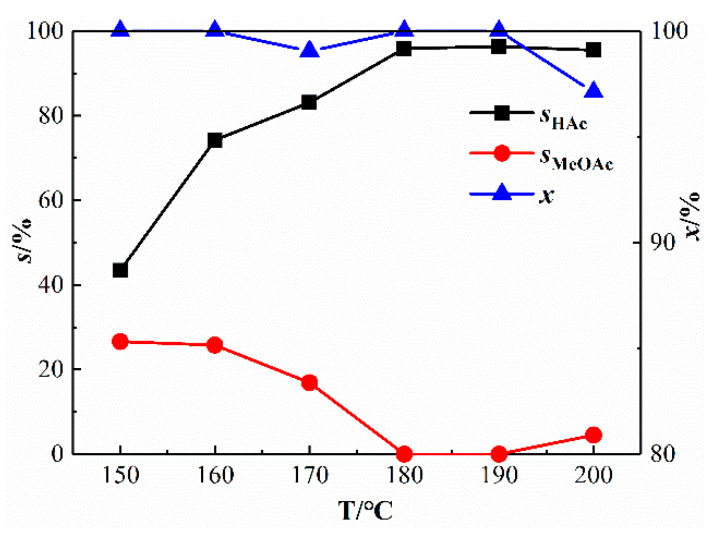
Catalytic performance of Rh(I)/Ru(III) bimetallic catalysts at different temperatures (*p* = 3.5 MPa; *t* = 60 min; w(CH_3_COOH) = 54 wt%; w(H_2_O) = 6 wt%; x/%: conversion rate of methanol).

**Table 1 materials-13-04026-t001:** Catalytic performance of Rh(I)/Ru(III), RuCl_3_·3H_2_O, and Rh_2_(CO)_4_Cl_2_ for methyl acetate carbonylation.

Catalyst	x/%	s_HAc_/%	y_HAc_/%	TOF/h^−1^	TON
Rh(I)/Ru(III)	94.9	85.2	80.9	3.2 × 10^7^	3.2 × 10^7^
Rh_2_(CO)_4_Cl_2_	90.3	49.6	44.8	7.6 × 10^6^	7.6 × 10^6^
RuCl_3_·3H_2_O	0	0	0	0	0

T = 190 °C; *p* = 3.5 MPa; *t* = 60 min; w(CH_3_COOH) = 54 wt%; w(H_2_O) = 6 wt%; x/%: Conversion rate of methanol; s_HAc_/%: the selectivity of acetic acid; y_HAc_/%: the yield of acetic acid; TOF: turnover frequency, i.e., the number of molecules formed per active site per second; TON: total number of products formed molecules per active site.

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
