# Peer review of "Study on Rh(I)/Ru(III) Bimetallic Catalyst Catalyzed Carbonylation of Methanol to Acetic Acid"

_materials, 2020, doi:10.3390/ma13184026_

Round 1
Reviewer 1 Report
The manuscript entitled "Study on Rh (I) -Ru (III) Bimetallic Catalyst Catalyzed Carbonylation of Methanol to Acetic Acid" reports a study in which a bimetallic catalyst Rh(I) -Ru(III) is synthesized and the structure by theoretical calculations has been studied. The performance of the catalyst in the methanol carbonylation reaction was evaluated and compared with the Montesanto catalyst. The authors state that the yield in acetic acid is high and that there is no formation of precipitate and only a small amount of water is produced .
The topic is consistent with the issues of the journal and could also be interesting for researchers working in this research area. However, I found the manuscript particularly difficult to follow and read. Many parts should be improved by trying to explain more clearly.
In the light of the above, I believe that the work in the current version cannot be considered for its publication, but a revision of it is necessary above all for what concerns the description of the operational activities and the results.
Here are some suggestions to the authors:
Paragraph materials and methods: the description of the used instruments is missing
Paragraph 2.2: more description should be given, for example briefly explaining Lanl2dz, B3LYP etc.
Paragraph 2.3.2: Although the molar ratio Rh2 (CO) 4Cl2 and RuCl3 3H2O is equal to 1: 2, but what is the actual quantity that is dissolved in 10 ml? What is the amount of the sodium carbonate solution that is added ?
Paragraph 2.3.3: This paragraph is not understandable. For example: Methyl acetate? RhI3 is used but is not described in section 2.1 as a reagent. This paragraph should be completely rewritten.
Figure 6: the caption should be more descriptive
Paragraph 3.3: for greater clarity, it is necessary to explain what x, s, y are
Author Response
Response to Reviewer 1 Comments
Thank you for your valuable revisions. The comments and suggestions are all useful for our research work and paper writing. According to the comments, we have revised the manuscript carefully, and gave our replies as follows (also marked with red color in our revised manuscript):
Point 1: Paragraph materials and methods: the description of the used instruments is missing 

Response 1: We are very sorry to make such a mistake. We have added it in revised manuscripts. (Line 98-123)
Point 2: Paragraph 2.2: more description should be given, for example briefly explaining Lanl2dz, B3LYP etc.
Response 2: Thank you for your careful reading of our manuscript. In this study, based on density functional theory (DFT), modern density functional methods are mainly based on the Kohn-Sham method, using b3lyp functionals. In 1994, Lee-Yang-Parr combined with Becke’s work and proposed the hybrid functional B3LYP. Currently, the Functional is the most extensive method for calculating organics containing heavy metals, and Lanl2dz is a pseudopotential basis set for errors in the calculation of heavy metals. Due to the limitation of the length of the article, the relevant documents are listed below for your reference.[1-4]
[1] Grimme, Stefan. Improved second-order Mller–Plesset perturbation theory by separate scaling of parallel- and antiparallel-spin pair correlation energies. The Journal of Chemical Physics, 2003, 118, 9095-9102.
[2] Zhao, Y.; González-García, Núria, Truhlar, D.G. Benchmark database of barrier heights for heavy atom transfer, nucleophilic substitution, association, and unimolecular reactions and its use to test theoretical methods. Journal of Physical Chemistry A, 2005, 109, 2012-2018.
[3] Cadierno, V.; García-Garrido, Sergio, E.; Gimeno, José, et al. Bis(allyl)—Ruthenium(IV) Complexes as Highly Efficient Catalysts for the Redox Isomerization of Allylic Alcohols into Carbonyl Compounds in Organic and Aqueous Media: Scope, Limitations, and Theoretical Analysis of the Mechanism. Journal of the American Chemical Society, 2006, 128, 1360-70.
[4] Jürgens, Barbara, Irran E , Senker, Jürgen, et al. Melem (2,5,8-triamino-tri-s-triazine), an important intermediate during condensation of melamine rings to graphitic carbon nitride: Synthesis, structure determination by X-ray powder diffractometry, solid-state NMR, and theoretical studies. Journal of the American Chemical Society, 2003, 125, 10288-10300.
Point 3 : Paragraph 2.3.2: Although the molar ratio Rh2 (CO)4Cl2 and RuCl3·3H2O is equal to 1: 2, but what is the actual quantity that is dissolved in 10 ml? What is the amount of the sodium carbonate solution that is added ?
Response 3: Thank you for your careful work. Rh2(CO)4Cl2 and RuCl3·3H2O are all dissolved in 10 ml water. The sodium carbonate solution added to the mixed solution is 50ml. We have corrected them in revised manuscripts. (Line 88)
Point 4 : Paragraph 2.3.3: This paragraph is not understandable. For example: Methyl acetate? RhI3 is used but is not described in section 2.1 as a reagent. This paragraph should be completely rewritten.
Response 4: We are very sorry to make such a mistake. We have corrected it in revised manuscripts. (Line 92-94)
Point 5 : Figure 6: the caption should be more descriptive.
Response 5: Thank you for your instructive suggestions. We all agree with your opinion. We have rewrited it in revised manuscripts.
Point 6 : Paragraph 3.3: for greater clarity, it is necessary to explain what x, s, y are
Response 6: Thank you for your careful reading of our manuscript. We have added it in revised manuscripts. x/%: Conversion rate of methanol; sHAc/%: the selectivity of acetic acid; yHAc/%: the yield of acetic acid. (Line 223-224)

Reviewer 2 Report
The present manuscript demonstrates the synthesis of Rh(I)-Ru(III) bimetallic catalyst by two-step synthesis (1) carbonylation of RhCl3 · 3H2O to form Rh2(CO)4Cl2 (2) followed by reaction with RuCl3 · 3H2O in the presence of Na2CO3. The confirmation of catalyst synthesis was done by XPS and FTIR. Interestingly, the developed Rh(I)-Ru(III) bimetallic catalyst displayed excellent performance in methanol carbonylation to afford acetic acid. The catalyst performance was higher than the Monsanto rhodium iodine [Rh(CO)2I2] catalyst (a benchmark catalyst used in industrial processes) and no deactivation was observed even at low CO pressure suggesting its commercial applicability. The finding in the manuscript is interesting and makes a significant contribution to the field. However, reporting such exotic results need extreme accuracy in term of reproducibility. There are certain flaws that need to be addressed prior to consideration for publication of the manuscript. I recommend major revisions. Some comments are:
- The introduction part can be improved by adding some other studies in the field and reporting the contribution of fermentation and biologically derived acetic acid etc.
- The FTIR spectra are not correct, The CO peaks are two broads, usually, we see two sharp peaks for metal carbonyls. Further, a comparison of compounds at each step should be added which will give information on how the insertion of CO changes the shape or position of the peaks. Some good reads are (Journal of industrial and engineering chemistry 61 (2018): 381-387)( Chem. 36 (1997) 3794)(Chem. Sci. 6 (2015) 3063)
- XPS interpretation is not good and a more detailed explanation should be added. Further did the author performed carbon correction on XPS data. Because of the charging effect usually, peak shifts are observed. Why there is a missing Ru3d5/2 peak for the Rh(I)-Ru(III)catalyst in Figure 6d.
- The main concern about the manuscript is the successful synthesis of Rh(I)-Ru(III)catalyst. FTIR and XPS are no confirmatory characterization tools. The author can use ESI-MS for the characterization or 13C NMR might help. Further, how sure is the author that the catalyst is pure, and the achieved activity is not because of other components. Usually in such kinds of bridged compounds dimerization is quite common. Sometime polymerization is also observed. Extensive column chromatography is used for the purification of such compounds from their dimers and polymers.
Author Response
Response to Reviewer 2 Comments
Thank you for your valuable revisions. The comments and suggestions are all useful for our research work and paper writing. According to the comments, we have revised the manuscript carefully, and gave our replies as follows (also marked with red color in our revised manuscript):
Point 1: The introduction part can be improved by adding some other studies in the field and reporting the contribution of fermentation and biologically derived acetic acid etc.
Response 1: Thank you for your instructive suggestions. We have added it in revised manuscripts. (Line 24-28)
Point 2: The FTIR spectra are not correct, The CO peaks are two broads, usually, we see two sharp peaks for metal carbonyls. Further, a comparison of compounds at each step should be added which will give information on how the insertion of CO changes the shape or position of the peaks. Some good reads are (Journal of industrial and engineering chemistry 61 (2018): 381-387)( Chem. 36 (1997) 3794)(Chem. Sci. 6 (2015) 3063)
Response 2: Thank you for your careful reading of our manuscript. We have rewrited it in revised manuscripts. Because the mechanism of the catalytic reaction of Rh(I) has been confirmed by both experiment and theory.[1,2] At present, how to terminate the reaction, separate and protect intermediates, and characterize (especially for rapid reactions) during the reaction process is still a common scientific problem. And rigorous quantum chemical calculations can achieve this goal. In the IR spectrum (Figure 1), Rh-CO of Rh (I) -Ru (III) bimetallic catalyst is located at 1600 cm-1 and 1700 cm-1. Comparing the infrared image of Rh2(CO)4Cl2, it can be seen that due to the influence of the ligand, the peak of the terminal carbonyl rhodium in the Rh(I)-Ru(III) bimetallic catalyst has a red shift.
Figure 1. FT-IR images of different catalyst (blue line: FT-IR image of Rh(I)-Ru(III) bimetallic catalyst; green line: FT-IR image of RhCl3·3H2O; red line: FT-IR image of RuCl3·3H2O; black line: FT-IR image of Rh2(CO)4Cl2)
[1] Lei, M.; Feng, W.L.; Hao, M.R.; Ji, Y.R.; Xu, Z.F. Theoretical study on the reaction of methanol carbonylation to acetic acid. Science in China (Chemistry Series B), 2001, 5, 462-467.
[2] Forster, D.; Dekleva, T.W. Catalysis of the carbonylation of alcohols to carboxylic acids including acetic acid synthesis from methanol. Journal of Chemical Education, 1986, 63, 204.
Point 3 : XPS interpretation is not good and a more detailed explanation should be added. Further did the author performed carbon correction on XPS data. Because of the charging effect usually, peak shifts are observed. Why there is a missing Ru3d5/2 peak for the Rh(I)-Ru(III) catalyst in Figure 6d.
Response 3: Thank you for your careful reading of our manuscript. Ru 3d5/2 is occupied by a pair of electrons, and no electron transfer occurs, so it is not reflected in XPS. Therefore, there is no Ru3d5/2 peak in the Rh(I)-Ru(III) bimetallic catalyst in Figure 7d.
Point 4 : The main concern about the manuscript is the successful synthesis of Rh(I)-Ru(III)catalyst. FTIR and XPS are no confirmatory characterization tools. The author can use ESI-MS for the characterization or 13C NMR might help. Further, how sure is the author that the catalyst is pure, and the achieved activity is not because of other components. Usually in such kinds of bridged compounds dimerization is quite common. Sometime polymerization is also observed. Extensive column chromatography is used for the purification of such compounds from their dimers and polymers.
Response 4: Thank you for your suggestion, we agree with you. Since the Rh(I)-Ru(III) catalyst has no skeleton C, 13C NMR cannot explain its structure. We have supplemented the characterization experiment and the ICP-OES results show that the mole ratio of Rh:Ru was obtained 1.15. Based on the characterization results of XPS and ICP-OES, we believe that the experimentally obtained Rh (I)-Ru (III) bimetallic catalyst configuration conforms to the theoretical design configuration. At the same time, we are preparing a single crystal of the catalyst, and obtaining the structure through single crystal diffraction experiments, which will finally prove the structure of the catalyst. This work is currently in progress. In addition, we provide the catalytic activity of Rh2(CO)4Cl2, RuCl3·3H2O, Rh(I)-Ru(III) bimetallic catalyst. The data shows that the bimetallic catalyst has a very obvious activity and selectivity improvement, which might be the result of the formation of a new catalyst. In addition, it can be known from the experimental results that RuCl3·3H2O has no catalytic activity. Therefore, it can be inferred that the active component of the catalyst is Rh(I).
Table 1 Comparison of catalytic effects between Rh(I)-Ru(III) bimetallic catalyst, Rh2(CO)4Cl2 and RuCl3·3H2O catalyst
|
Catalyst system |
x(Methanol)/% |
s(Acetic acid)/% |
|
Rh(I)-Ru(III) |
100 |
96.32 |
|
Rh2(CO)4Cl2 |
90.3 |
49.6 |
|
RuCl3·3H2O |
0 |
0 |
T=190℃,p=3.5MPa,t=60min,ω(CH3COOH)=54%,ω(H2O)=6%

Reviewer 3 Report
The authors investigated carbonylation of acetic acid to methanol using a synthesized Rh-Ru bimetallic catalyst. This reaction in general is important for a variety of industries and the groups findings are interesting. However, there are several missing characterizations of the catalyst they produced as well as reactions that many readers will consider necessary. At this point, I cannot recommend this manuscript for publication. Here are a few issues I found with the manuscript:
- The English needs to be revised
- In line 95 it states that Figure 1 shows the distance between the central Rh-Ru atoms is 0.2957 nm, however, this isn’t shown in the actual figure
- Line 104-105 claims mechanism in Figure 3 is similar to Monsanto’s catalyst w/o showing Monsanto’s mechanism.
- Figure 4 shows the free energy values between barriers, however, it doesn’t state the difference in free energy between states. Please add relative free energy at each state.
- Figure 5, please show IR of starting materials (RhCl3, Rh2(CO)4Cl2 and RuCl3) for comparison
- How many times was the reaction in Table 1 reproduced? Also acronyms need to be defined.
- Figures 7-10, please show comparison of Monsanto’s catalyst
- How stable is the Rh-Ru catalyst. Have you calculated the Turnover Numbers as well as Turnover Frequency? This should be considered essential when claiming catalyst superiority over an established catalyst.
This should not be considered a comprehensive list of edits that need to be done but instead a guidance before future submission. The work done is interesting and I would like a chance to read it again when it’s ready.
Author Response
Response to Reviewer 3 Comments
Thank you for your valuable revisions. The comments and suggestions are all useful for our research work and paper writing. According to the comments, we have revised the manuscript carefully, and gave our replies as follows (also marked with red color in our revised manuscript):
Point 1: The English needs to be revised
Response 1: Thank you for your suggestion, it is very important to us. We have proofread this article by a native English speaker.
Point 2: In line 95 it states that Figure 1 shows the distance between the central Rh-Ru atoms is 0.2957 nm, however, this isn’t shown in the actual figure
Response 2: We are very sorry to make such a mistake. We have corrected it in the revised draft. In addition, we use angstroms as the unit of bond length. Therefore, 0.2957 nm is corrected to 3.5846 angstroms.
Point 3 : Line 104-105 claims mechanism in Figure 3 is similar to Monsanto’s catalyst w/o showing Monsanto’s mechanism.
Response 3: Thank you for your suggestions. We have added it in revised manuscripts. (Line 136-137)
Point 4 : Figure 4 shows the free energy values between barriers, however, it doesn’t state the difference in free energy between states. Please add relative free energy at each state.
Response 4: Thank you for your suggestions. We have corrected it in revised manuscripts. (Line 170 Figure 5)
Point 5 : Figure 5, please show IR of starting materials (RhCl3, Rh2(CO)4Cl2 and RuCl3) for comparison
Response 5: Thank you for your instructive suggestions. We have added it in revised manuscripts. (Figure 6)
Point 6 : How many times was the reaction in Table 1 reproduced? Also acronyms need to be defined.
Response 6: Thank you for your careful work. The reactions in Table 1 are repeated once, and the data obtained is reliable. We have added it in revised manuscripts.
Point 7 : Figures 7-10, please show comparison of Monsanto’s catalyst
Response 7: Thank you for your careful reading of our manuscript. We compared the reactivity of Rh(I)-Ru(III) bimetallic catalysts, RuCl3·3H2O and Rh2(CO)4Cl2 to catalyze carbonylation to acetic acid. It can be seen from Table 1 that under the same conditions, the selectivity and yield of the Rh(I)-Ru(III) bimetallic catalyst are significantly higher than that of the Monsanto catalyst. Therefore, we did not repeat the performance evaluation experiment of the catalytic activity of Monsanto catalyst under other conditions. We only did optimization experiments on bimetallic catalysts.
Table 1. Catalytic performance of Rh(I)-Ru(III), RuCl3·3H2O and Rh2(CO)4Cl2 for methyl acetate carbonylation
|
Catalyst |
x/% |
sHAc/% |
yHAc/% |
|
Rh(I)-Ru(III) |
94.9 |
85.2 |
80.9 |
|
Rh2(CO)4Cl2 |
90.3 |
49.6 |
44.8 |
|
RuCl3·3H2O |
0 |
0 |
0 |
T=190℃; p=3.5MPa; t=60min; w(CH3COOH)=54% wt; w(H2O)=6% wt; x/%: Conversion rate of methanol; sHAc/%: the selectivity of acetic acid; yHAc/%: the yield of acetic acid
Point 8 : How stable is the Rh-Ru catalyst. Have you calculated the Turnover Numbers as well as Turnover Frequency? This should be considered essential when claiming catalyst superiority over an established catalyst.
Response 8: Your suggestions are very important, and this will be an important part of our next work. Because the catalyst is a homogeneous catalytic reaction carried out in a batch reactor. The biggest problem of industrial catalysts is that after the reaction, the partial pressure of CO decreases, and the precipitation of Rh(â… ) is subsequently deactivated. Existing research mainly judges the inactivation of Rh by whether there is precipitation in the reactor after the reaction, and there is no precipitation of Rh in this research. Since the corresponding industrialization experiment has not been carried out, we will put forward relevant data in the follow-up report.
This should not be considered a comprehensive list of edits that need to be done but instead a guidance before future submission. The work done is interesting and I would like a chance to read it again when it’s ready.
Thank you very much for your great efforts on our manuscript. Your suggestion makes our manuscript more perfect. Thank you for your recognition of our work, we will continue to work hard.

Reviewer 4 Report
The reviewer comments are attached as pdf.

Author Response
Response to Reviewer 4 Comments
Thank you for your valuable revisions. The comments and suggestions are all useful for our research work and paper writing. According to the comments, we have revised the manuscript carefully, and gave our replies as follows (also marked with red color in our revised manuscript):
Point 1: The first reservation in the experimental part is the lack of information on techniques already used. The author should describe in enough detail how they collected the XPS spectra (apparatus, energy, source…etc) and infrared spectra (the type of spectrometer, what technique was applied? DRIFTS or what, how was this data was evaluated?).
Response 1: Thank you for your careful work. This is a great help to perfect our article. We have added it in revised manuscripts. (Line 98-123)
Point 2: My second concern about the experimental part (characterization) is the lack of basic
characterizations. Here I mean the elemental analysis in particular which can give you the formulate weight of the resulting complex. Authors should characterize their catalyst with ICP-OPEs, Atomic absorption spectrometry (AAS), or EDX analysis. After that, they have to determine the experimental ratio of elements (Ru: Rh: Cl). This is imperative to decide the correct structure. Additionally, BET surface area of the catalyst and XRD pattern should be collected, especially this is new material
Response 2: Thank you for your instructive suggestions. We have corrected it in revised manuscripts. We use homogeneous catalysts, which exist as molecules in the reaction system. Since our catalyst is a homogeneous catalyst, the BET specific surface area of the catalyst has no direct relationship with performance evaluation. We have supplemented the characterization experiment and the ICP-OES results show that the mole ratio of Rh:Ru was obtained 1.15. Based on the characterization results of XPS and ICP-OES, we believe that the experimentally obtained Rh (I)-Ru (III) bimetallic catalyst configuration conforms to the theoretical design configuration. For XRD, we are trying to cultivate a single crystal of the catalyst, hoping to obtain the result of the single crystal diffractometer, and thus obtain a clear structure of the catalyst, which will be reported in the subsequent research results.
Point 3 : In section 2.3.3 where you talk about the catalytic testing, I am missing the method you quantified reactants and reaction products. In other words, how did you analyze the reaction mixture? Gas chromatography …etc. This has to be explained in detail in the experimental section.
Response 3: Thank you for your suggestion, it is very important to us. We have added section 2.5 to the manuscript to illustrate our method of analyzing products.
Point 4 : First of all, I suggest moving the whole section about the mechanism (DFT calculations) back at the end of the paper after showing the catalyst characterization (XPS, DRIFTS, and the other requested measurements for evaluation). It makes more sense to discuss the mechanism after presenting the structure and activity.
Response 4: Thank you for your suggestions. The catalyst is based on existing work (we have published documents), and theoretical calculations are performed on the designed molecular configuration to obtain the reaction mechanism. The reaction energy barrier diagram obtained from the calculation results shows that the rate-determining step barrier of the catalytic reaction of the catalyst is lower than that of the rhodium-iodide catalyst. Comprehensive characterization and performance evaluation results confirmed the feasibility of the theoretical design, thus providing a feasible way for catalyst design. Therefore, we believe that theoretical calculations should be placed at the front of the paper. Hope the review experts understand.
Point 5 : In section 3.1 the discussion about Ru-Ru bond distance is confusing. First you mention 0.2957 nm for planar structure then mention the 0.3683 nm is calculated for planar structure. I do not know where the first distance came from. Please clarify that.
Response 5: We are sorry for making such a mistake. We have corrected it in the revised draft. In addition, we use angstroms as the unit of bond length. Therefore, 0.2957 nm is corrected to 3.5846 angstroms. The Rh-Ru bimetallic catalyst in this paper is a four-coordinate dimer structure. From the perspective of electronic steric hindrance, the planar structure should be the structure with the least steric hindrance and the lowest energy, but the most stable structure obtained by calculation which is a folded structure, so we compared the planar structure of the catalyst with fictitious calculations. For the two configurations, the biggest change is that the distance between Rh-Ru is close, which indicates that there is a certain interaction between Rh-Ru before. It offsets the increase in the energy of the catalyst structure brought about by the folded structure, so that the overall energy of the catalyst is reduced, which is more stable.
Point 6 : Figure 2 is just mentioned without enough explanation which is not of any help for the reader. Either to explain in detail or remove.
Response 6: Thank you for your guiding comments. We have added a diagram of the catalytic mechanism of Monsanto catalyst. It is used to show that our reaction mechanism is similar to that of Monsanto catalyst. At the same time, it shows that the catalytic mechanism is divided into four steps, and finally it is reduced to a catalyst by hydrolysis.
Point 7 : For Fig. 5 you should indicate the Y-axis (Intensity as absorbance, transmission or diffuse reflectance?
Response 7: Thank you for your careful work. We have marked the Y axis of the infrared chart as absorbance. (Figure 6)
Point 8 : How were XP spectra calibrated? The energy calibration for all energy regions used should be indicated.
Response 8: Thank you for your careful reading of our manuscript. We have added it in revised manuscripts. XPS requires calibration since charging of the sample will influence the kinetic energies. Spectra from samples have been charge corrected to give the adventitious C1s spectral component (C–C, C–H) binding energy of 284.8 eV. This process has an associated error of ± 0.1– 0.2 eV. (Line 186-189)
Point 9 : Minor:
-Bod distances are normally presented with the unit Angstrom instead of nanometer (nm) which is typical for particle sizes.
-Figure captions should give enough information.
Response 9: Thank you for your suggestion, it is very important to us. We have corrected it in revised manuscripts and enriched graphic title information.

Round 2
Reviewer 1 Report
I believe that the manuscript in the present version can be accepted for its publication.
With best regards
Author Response
Response to Reviewer 1 Comments
Thank you for your valuable suggestions on our manuscript. And these suggestions make our manuscript perfect. Sincerely, thank you for your support to our work, we will work harder.
With best regards

Reviewer 2 Report
I carefully re-evaluated the author's response to the reviewer's comments. The suggested change has been incorporated and the manuscript quality has been improved significantly. I recommend publication of the manuscript in its current format.
Author Response
Response to Reviewer 2 Comments
Thank you for your valuable suggestions on our manuscript. And these suggestions make our manuscript perfect. Sincerely, thank you for your support to our work, we will work harder.
With best regards
Reviewer 3 Report
Thank you for addressing most of my concerns regarding your manuscript. However, some of the new figures are behind the old figures; see figures 2 and 5) Also, the caption for figure 4 isn’t near the actual figure. I assume this is a simple error that can readily be fixed. From what I can see, it looks like the new figures are an improvement. However Figure 5 is completely covered by the old figures. Regarding catalyst stability, I understand the issue regarding catalyst precipitation, however, my question regarding TON/TOF was never answered. I looked through your paper so I could calculate the TON myself, however, it only mentioned that 0.03 g of Rh-Ru was added. Did you add the same number of moles of the other two catalysts in table 1? Doing a quick rough calculation, I got approx. 3000 TON suggesting the TOF should be at least 3,000 per h. Is this correct?
At this point, I believe another less extensive revision is still a required. I look forward to reviewing this paper again.
Author Response
Response to Reviewer 3 Comments
Thank you for your valuable revisions. The comments and suggestions are all useful for our research work and paper writing. According to the comments, we have revised the manuscript carefully, and gave our replies as follows (also marked with red color in our revised manuscript):
Point 1: Thank you for addressing most of my concerns regarding your manuscript. However, some of the new figures are behind the old figures; see figures 2 and 5) Also, the caption for figure 4 isn’t near the actual figure. I assume this is a simple error that can readily be fixed. From what I can see, it looks like the new figures are an improvement. However Figure 5 is completely covered by the old figures. Regarding catalyst stability, I understand the issue regarding catalyst precipitation, however, my question regarding TON/TOF was never answered. I looked through your paper so I could calculate the TON myself, however, it only mentioned that 0.03 g of Rh-Ru was added. Did you add the same number of moles of the other two catalysts in table 1? Doing a quick rough calculation, I got approx. 3000 TON suggesting the TOF should be at least 3,000 per h. Is this correct?
At this point, I believe another less extensive revision is still a required. I look forward to reviewing this paper again.
Response 1: Thank you for your careful work. This is a great help to perfect our article. we have carefully examined and revised all figures in Figures 2, 5 and 4 in the revised manuscript. And we added the same number of moles of the other two catalysts in table 1. Through our calculation, the TOF of Rh (I)-Ru (III) catalyst is 3.2*10^7 per h, and the TOF of Rh2(CO)4Cl2 catalyst is 7.6*10^6 per h. We have added it in revised manuscripts. (Line 118-119, 222-226)
Table 1. Catalytic performance of Rh(I)-Ru(III), RuCl3·3H2O and Rh2(CO)4Cl2 for methyl acetate carbonylation
|
Catalyst |
x/% |
sHAc/% |
yHAc/% |
TOF/ h-1 |
TON |
|
Rh(I)-Ru(III) |
94.9 |
85.2 |
80.9 |
3.2*10^7 |
3.2*10^7 |
|
Rh2(CO)4Cl2 |
90.3 |
49.6 |
44.8 |
7.6*10^6 |
7.6*10^6 |
|
RuCl3·3H2O |
0 |
0 |
0 |
0 |
0 |
T=190℃; p=3.5MPa; t=60min; w(CH3COOH)=54% wt; w(H2O)=6% wt; x/%: Conversion rate of methanol; sHAc/%: the selectivity of acetic acid; yHAc/%: the yield of acetic acid; TOF: turnover frequency is the number of molecules formed per active site per second; TON: total number of products formed molecules per active site
TOF = reaction times /(number of active sites× reaction time)
TON= reaction times /number of active sites

Reviewer 4 Report
The authors improved the manuscript significantly and I suggest publication after only one minor correction in Figure 6:
IN the caption they wrote ''FT IR images''. This is wrong and FTIR FTIR is described as Spectra / Spectrum (please correct). It should be corrected to FTIR spectra
Author Response
Response to Reviewer 4 Comments
Thank you for your valuable revisions. The comments and suggestions are all useful for our research work and paper writing. According to the comments, we have revised the manuscript carefully, and gave our replies as follows (also marked with red color in our revised manuscript):
Point 1: The authors improved the manuscript significantly and I suggest publication after only one minor correction in Figure 6: IN the caption they wrote ''FT IR images''. This is wrong and FTIR FTIR is described as Spectra / Spectrum (please correct). It should be corrected to FTIR spectra
Response 1: Thank you for your careful reading of our manuscript. We are very sorry to make such a mistake. We have corrected it in revised manuscripts. (Line 185-186)
